# Management Decisions: The Effectiveness and Size of the Emergency Medical Team

**DOI:** 10.3390/ijerph19073753

**Published:** 2022-03-22

**Authors:** Marlena Robakowska, Daniel Ślęzak, Przemysław Żuratyński, Kamil Krzyżanowski, Anna Tyrańska-Fobke, Magdalena Błażek, Jarosław Woroń

**Affiliations:** 1Department of Public Health & Social Medicine, Medical University of Gdańsk, 80-210 Gdańsk, Poland; mrobakowska@gumed.edu.pl (M.R.); anna.tyranska-fobke@gumed.edu.pl (A.T.-F.); 2Department of Medical Rescue, Medical University of Gdańsk, 80-210 Gdańsk, Poland; przemyslaw.zuratynski@gumed.edu.pl (P.Ż.); kamil.krzyzanowski@gumed.edu.pl (K.K.); 3Division of Quality of Life Research, Medical University of Gdańsk, 80-210 Gdańsk, Poland; magdalena.blazek@gumed.edu.pl; 4Department of Clinical Pharmacology, Jagiellonian University, 31-531 Kraków, Poland; j.woron@uj.edu.pl

**Keywords:** medical emergency teams, paramedics, Poland

## Abstract

In Poland, often for economic reasons, the staffing of medical rescue teams is limited to the legally required minimum. This gives rise to problems related to the effectiveness and efficiency of medical rescue teams. A literature review did not find any sources addressing the issue of the verification of the effectiveness of paramedic teams depending on the personnel composition of units. The aim of the study was to analyze the effectiveness of resuscitation depending on the size of the medical rescue team, comparing the work of two- and three-person teams. In total, 100 two-person teams and an analogous number of three-person units were studied. Statistical analyses were performed using the IBM SPSS Statistics 24 package. The results showed that the assessment of the condition of the victim as well as the ability to assess the heart rhythm and monitor the condition during advanced measures were more effective in three-person teams; three-person teams also used oxygen more frequently during advanced life support (ALS). Most of the elements influenced the quality of resuscitation and it can be unequivocally stated that the work of three rescuers is more efficient and definitely more effective.

## 1. Introduction

Legal acts currently in force in Poland define the types of units providing medical rescue services and divide them into basic (P) and specialist (S) teams. These regulations indicate, among other things, the required personnel composition of each team and, more precisely, the minimum number of people required to undertake the medical actions. The legislator has indicated that basic teams are staffed by at least two paramedics and/or nurses who are authorized to undertake medical rescue actions [1]. The law regulates only the minimum composition of a medical rescue team. Unfortunately, this is often due to economic reasons. There are entities that extend the team with a third member (e.g., the operator of emergency medical services in Elbląg and Olsztyn); however, due to a lack of additional funds from the National Health Fund (NHF), such practices are rare.

For over a dozen years, diagrams of procedures—including recommendations for the resuscitation of adults—have been developed and published by the European Resuscitation Council (ERC) [2]. In Poland, since 2001, the activities in this field on behalf of the ERC have been carried out by the Polish Resuscitation Council (PRR) [3]. Every five years, publications containing recommendations for medical rescue operations in life-threatening situations are successively published. For the purposes of this study, the presented scope of topics was deliberately limited only to the algorithm of advanced life support in adults.

The choice of the procedure is a prerequisite for obtaining a recording of the heart rate. It is one of the key steps in assessing the condition of the victim. It allows you to make a decision on the choice of an algorithm conditioning the performance of defibrillation or one based on pharmacotherapy and basic life support (BLS) procedures.

Ventricular fibrillation (VF) and pulseless ventricular tachycardia (VT) are rhythms requiring immediate defibrillation. There should be a 2 min interval between each subsequent rhythm evaluation and any shock load.

Pharmacotherapy is usually limited to two preparations, adrenaline and amiodarone, and its beginning is determined by the time of the third defibrillation [4]. Early drug administration is not normally recommended. The bases of this algorithm path are BLS and defibrillation; pharmacotherapy takes a further place in the presented protocol.

Asystole and pulseless electrical activity (PEA) constitute the second path of the standard where BLS is recommended along with the fastest possible administration of 1 mg of adrenaline [5]. In this case, pharmacotherapy, in addition to chest compressions and ventilation, forms the basis of the algorithm and defibrillation is contraindicated. The introduction of other routine preparations is not recommended. As with the timing of the discharge in defibrillation rhythms, the administration of the drug in this case determines the appropriate time interval. Currently, it is 3–5 min [6].

The aim of the study was to analyze the effectiveness of resuscitation depending on the size of the medical rescue team, comparing the work of two- and three-person teams.

## 2. Materials and Methods

A survey was conducted on a group of certified active paramedics. The participants of the study were recruited from among the co-workers of the authors employed in ambulance stations and hospital emergency departments in the Pomeranian Voivodeship. The assumptions of the study as well as the significance of the obtained results for clinical and organizational practice were explained to the participants. The competence, scope of authority to perform rescue procedures and administer medications among all respondents were the same. According to the current law, every paramedic in Poland has the same qualifications. Job seniority and type of school attended do not affect the scope of competence. None of the components indicated above constituted a statistical difference and did not affect the final result of the study.

The study assessed the quality of life-saving procedures used and the time taken to perform them according to the composition of the teams. The assessment of basic vital functions, defibrillation technique (electrotherapy-defibrillation rhythms), chest massage (breaks in chest compressions, chest compression depth, frequency of chest compressions), oxygen therapy, pharmacotherapy and drug dosage were verified. The aspects subject to verification were assessed using the evaluation card of the authors, which took into account the correctness and execution time of the individual elements of the algorithm. All observations were recorded on the evaluation card. The advanced life support (ALS) phantoms of the AMBU company were used with a computer analysis that allowed the continuous quality control of the actions performed and ensured that the equipment was compatible with the equipment of basic medical rescue teams. The test conditions were the same for each team. The room was closed to prevent the task from being interrupted by random people. The lighting and thermal conditions were similar in each case. Before starting the task, all participants drew a scenario from the group of defibrillation and non-defibrillation rhythms. They were given time to familiarize themselves with the assumptions of the task. The prepared simulations differed in the description of the surroundings (bus stop, staircase, pavement, etc.) whereas the elements concerning the condition of the injured person were unchanged (state of consciousness, quality of breathing and circulatory symptoms). None of the prepared scenarios covered the subject of special conditions or threats resulting from the surrounding environment. They relied solely on a universal algorithm for managing adult cardiac arrest. Each team had time before starting the task to prepare and arm, at their own discretion, the available equipment located in the rescue bags and backpacks. Access was also provided to several models of defibrillators of the most popular brands (ZOLL M-series, ZOLL E-series, Lifepak 12, Lifepak 15). The time to perform the procedure was limited to 10 min. After its expiration, the evaluation was discontinued, regardless of the severity of the procedures performed. Statistical analyses were performed using the IBM SPSS Statistics 24 package to answer the research questions posed. It was used to analyze the basic descriptive statistics and Student’s *t*-tests for the independent samples as well as Mann–Whitney tests and chi-squared tests. A *p* < 0.05 was used as the level of significance in this chapter. A *p* < 0.1 level was considered to be significant for the level of the statistical trend.

## 3. Results

A total of 463 individuals participated in the study. The participants were divided into teams of two and three. The study was voluntary and anonymous. The age range of the participants was 26–48 years; 22.5% (104 individuals) of the respondents were female and the remaining 77.5% (359 individuals) were male. A total of 100 teams of 2 people and the same number of 3-person teams participated in the study. Among the teams of two, 50 cardiac arrest scenarios with shockable rhythms and 50 with non-shockable rhythms were randomly drawn. The same procedure was carried out in teams of three. A randomly selected group of participants took part in both the scenarios of 2-person teams and the scenarios of 3-person teams. Shorter compression intervals applied to the 3-person teams. Therefore, these teams could be considered to be more effective in chest compressions compared with the 2-person teams (Table 1). The correct depth value was obtained by the 3-person teams and they also had a value closest to the desired range of the ERC guidelines, which were described in the Introduction. Using the Student’s *t*-test for the independent samples, we examined whether the 2- and 3-person teams differed in the mean depth of chest compressions. The result of the performed analysis was statistically significant; thus, the compared groups differed in this respect (Table 1).

The results in both study groups were comparable. It was indicated that the recommended 120 compressions/min were exceeded in each group (Table 1). Using the Student’s *t*-test for the independent samples, it was proven that the 2-person teams did not differ from the 3-person teams in terms of chest compression frequency.

The 3-person teams were more effective during electrotherapy (Table 1). The results indicated that the 3-person teams administered the first dose of epinephrine significantly faster than the 2-person teams (Table 1).

The Student’s *t*-test for the independent samples showed that these differences were statistically significant. According to the percentage distributions, the 3-person teams were more efficient in assessing the heart rhythm and current status by performing analyses every 2 min (Table 1).

The Student’s *t*-test for the independent samples was also used to test whether the groups differed in terms of the time taken to administer oxygen therapy. The result of the test proved to be statistically insignificant; thus, both groups started oxygen therapy at similar times (Table 1).

To investigate which group of subjects was more effective in chest compressions, a Student’s *t*-test for the independent samples was performed. This resulted in statistically significant results, indicating that there were differences in the chest compression intervals. The Cohen’s *d* measure of the strength of the effect of the differences showed that the differences were quite strong (Figure 1).

The 3-person teams were found to use gel more often compared with the 2-person teams; the 3-person teams delayed defibrillation significantly less often and performed a rhythm analysis and discharge more often (Table 2).

For the first discharge time, the Student’s *t*-test for the independent samples showed that the compared groups differed. A statistically significant shorter time characterized the 3-person teams compared with the 2-person teams. According to the measure of the strength of the effect of differences, these differences were strong.

Statistically significant differences were also obtained when measuring the timing of the first dose of epinephrine. When analyzing the percentage distribution, it could be seen that the 3-person teams were significantly more likely to use correct dosing compared with the 2-person teams. However, there was no significant relationship between the correct dosing and the team groups (Table 2).

The analysis showed that significantly faster rhythm recordings were obtained by the 3-person teams compared with the 2-person teams. The chi-squared test proved that there were statistically significant relationships between the groups and the quick look measurement and analysis every 2 min. The Student’s *t*-test for the independent samples was used to test whether the teams differed statistically significantly in terms of the time to obtain the rhythm recording (Table 2). The 3-person teams were significantly more likely to use oxygen therapy than the 2-person teams (Table 2). This result was obtained using the chi-squared test of independence.

## 4. Discussion

During the analysis of the literature, no sources dealing with the issues related to the verification of the effectiveness of work in rescue teams depending on the personnel composition of the units were noticed. The authors of numerous publications list and analyze the elements that may have a potential impact on obtaining a return of spontaneous circulation (ROSC), but do not discuss the impact of the number of rescuers on the quality of operations. The discussion on the impact of the numerical composition of the team on the possibility of obtaining an ROSC could be based on the analysis and comparison of the effectiveness of both units in individual components that ultimately constitute the overall resuscitation and the opinion of specialists regarding their impact on survival after cardiac arrest. 

The authors of the guidelines of the ERC explicitly indicate that immediate help should be summoned if cardiac arrest is observed [6]. These recommendations concern both untrained and qualified rescuers. In the first minutes of action, many rescue measures have to be quickly implemented. It is obvious that the number of people, in addition to their qualifications and experience, is crucial. Often, during resuscitation or after an ROSC, the competence of the rescuers is not sufficient to provide further specialist care. Several rescue techniques are not allowed without medical supervision. The use of muscle relaxants, pressor amines and anesthetics exceeds the competence of intermediate personnel. The evacuation of the casualty for follow-up care within the intensive care unit is another factor indicating that the number of rescuers can have a positive impact on the work rate. The mass of the casualty and the volume of the room space as well as the number and weight of the equipment used to support vital functions make preparation for transport energy-consuming and time-consuming. In this study, the authors showed that, among both groups of teams, it was the three-person units that were quicker to ask other teams for support and thus were quicker to receive the expected help. The position of the authors on the need to immediately call for help is the same as that of the ERC specialists.

Recommendations for minimizing interruptions in sternal compressions are found repeatedly in the literature. They emphasize that the compression time should be maintained over 60% of the entire resuscitation cycle. The maximum pause should not exceed 10 s [7,8,9,10,11]. Prolonged pauses in sternal compression can be fatal to spontaneous circulation [12]. The time limits were rarely exceeded by the rescuers in the three-person units. The care to minimize interruptions was probably associated with more personnel, who were able to devote time to implement other procedures. Among the two-person teams, the lack of additional team member(s) was noticeable. Only a few team members were able to cope with limiting interruptions and acting in accordance with the algorithm, but these were usually individuals who had been working together for a long time in real-world settings and who regularly updated their skills and knowledge. It should be noted that the indicated element of ALS/BLS could be maintained at a high level among both groups of teams, but frequent training with discussions about the most essential elements of the algorithm conditioning the ROSC is necessary.

An increase in the resuscitation efficacy was observed when the depth of sternal compression was within 4.5–5.5 cm [13,14]. At present, the recommendations of international organizations such as the International Liaison Committee on Resuscitation (ILCOR) and the ERC indicate that, during resuscitation, chest compressions should aim for a depth of about 5 cm and not exceeding 6 cm [15]. During the study, both groups of teams did not reach the 5 cm limit. The differences were relatively small, with a discrete advantage for the three-person units. It could be assumed that, regarding the above element, the performance of the studied units was comparable and in need of improvement. There was a tendency to implement advanced techniques such as intubation and intravascular access at the expense of a proper quality of sternal compression. It is worth emphasizing that this is the element to which special attention should be paid, both during the initial and the postgraduate teaching process.

Idris et al., in study results published in 2012 and 2015, showed an increase in the survival of patients with a treated cardiac arrest when a sternal compression rate of 100–120/min was used during the therapy [16,17]. The authors pointed out that overestimating the frequency often results in a decrease in the compression depth and a significant increase in rescuer fatigue, thus making the effectiveness of the therapy decrease [16,17]. The analysis of our study results showed that both groups of teams maintained a frequency slightly above 120 compressions/min. The two-person units were found to be closer to the pattern; however, similar to the depth of sternal compression, a system improving the teaching of both BLS elements should be implemented.

A rapid rhythm analysis can effectively treat arrhythmias that require defibrillation. The chances of the restoration of circulation during ventricular fibrillation (VF)/ventricular tachycardia (VT) treatments correlate closely with the elapsed time and quality of BLS [18,19]. Blom et al., in their paper published in 2014 concerning the dependence of defibrillation effectiveness on the time of the procedure, pointed out the significant importance of the discharge in the first minutes after cardiac arrest [20]. An efficacy of 50–70% was demonstrated for defibrillation performed within the first 3–5 min after the mechanism of cardiac cessation was recognized [21,22]. The optimal solution during the initial assessment of the condition of the victim was a two-track procedure performed simultaneously by two rescuers. One rescuer focused on the assessment of the vital functions; the other rescuer verified the recording of the heart rhythm. An electrocardiogram (ECG) analysis with simultaneous defibrillator charging is an option worth considering. If VF/VT is observed, immediate defibrillation is possible. With reference to the opinions of the authors indicated above, performing a shock as soon as possible after the observation of a defibrillation rhythm may have a crucial influence on the treatment outcome. In the first stage of operations, there is no need to use a lead to monitor the limb leads. Due to the time required to properly attach the electrodes to the chest, this may actually delay the decision to require defibrillation. A quicker, simpler solution is to use the quick look rhythm analysis technique with the defibrillator paddles. This function is standard on the apparatus that rescue teams are equipped with. At further stages of resuscitation, the use of multifunctional leads [6] or monitoring with limb leads is recommended. During this study, the quick look technique was observed relatively rarely during the first analysis of the heart rhythm. The rescuers noticeably delayed defibrillation, focusing on the installation of leads for conventional heart rate monitoring. This begs the question, why do team members not remember the capabilities of their everyday devices? As previously mentioned, when discussing the results of this experiment, the advantage in this aspect was gained by the three-person teams, obtaining shorter times to the first discharge. It should be mentioned that the results obtained were not ideal. According to the literature, a quick analysis of the heart rhythm is an element potentially influencing the moment of defibrillation and thus the possibility of obtaining the ROSC. Special attention should be paid to this in the teaching process.

Another item tested was the ability to perform a rhythm analysis at 2 min intervals. The recommendations on which the assessment was based can be found in the ERC Guidelines, 2015 [6]. In the indicated aspect, the three-person teams prevailed again. An analogy could be seen here with the outcome of time control during the use and supervision of pharmacotherapy. The data indicated that the interventions were more effective when a greater number of personnel were present.

The effectiveness of electrotherapy is influenced by minimizing interruptions in sternal compression during the device preparation and removing the delay in the discharge decision when defibrillatory rhythms are identified [6]. The discharge efficacy may be negatively affected even by 5–10 s pauses in BLS [7,8,23,24]. Another element affecting the quality of resuscitation in the case of defibrillation rhythm management is energy grading. If there is no positive effect, it is recommended that the power is gradually increased to reach the maximum value by the third discharge [25]. Maintaining one energy level throughout the resuscitation may not have the desired effect and may further impair the myocardial cells. The use of multifunctional electrodes [6] or the use of defibrillation gel is suggested to reduce the resistance and improve the defibrillation efficiency. The analysis of the electrotherapy quality results, referring to the opinions and guidelines of specialists, indicated that the knowledge of the principles of defibrillation treatment was observably higher among the three-person teams. Their care of individual elements affecting the effectiveness of the procedure was rated higher in comparison with the two-person units.

Defibrillation rhythms should be treated with adrenaline and amiodarone. The time of administration of the first doses of drugs determines the moment of the third defibrillation. In case the discharge does not lead to an ROSC, adrenaline and amiodarone have been introduced into the standard [26]. Currently, the recommended dosage of adrenaline limits a single application to 1 mg administered at 3–5 min intervals [27]; amiodarone is limited to 300 mg diluted in 5% glucose [6]. During the treatment of non-defibrillation rhythms, only adrenaline can be found in the algorithm; other drugs are not used as a standard [6]. The results indicated that the three-person teams administered the first dose of adrenaline significantly faster than the two-person teams. Statistically significant differences were also obtained when the time of administration of the first dose of adrenaline was measured. Analyzing the percentage distribution, it was proven that the three-person teams used the appropriate dosage significantly more often compared with the two-person teams. On the other hand, there was no significant correlation between the proper dosing and the team groups.

According to the current standards, medical oxygen should be administered from the onset of therapy in such a way as to achieve its high concentration in the administered breathing mixture as quickly as possible [6]. According to the literature findings, oxygen supplementation during ALS may lead to more frequent ROSCs [28]. The element distinguishing the teams with a greater number of personnel was oxygen therapy. In such cases, ventilation enriched with the respiratory mixture was used more frequently. Therefore, in the aspect mentioned, the effectiveness of the three-person teams was considered to be higher.

Summarizing all the elements influencing the quality of resuscitation, it can be unequivocally stated that the work of three rescuers was more efficient and definitely more effective. It was noted that in single, isolated elements, the level of both groups was similar; however, in relation to the greater number of procedures imposed by the ERC, the three-person units were more effective. This does not mean, however, that the obtained results were at a satisfactory level and inspired optimism. 

The study showed that in individual aspects of the study there were differences between the teams of two people and the teams of three people, in favor of the teams consisting of three people. However, this was not an unequivocal advantage of the three-person teams, which would justify the obligatory extension of the teams to three people. Practically speaking, the increase in the number of team members from two to three is probably very low. The widespread use of demonstrating in a system of two-person teams is a type of compromise, taking into account the satisfactory quality of services provided and the costs incurred.

According to the research, nearly 80% of the costs of maintaining medical rescue teams in Poland are the costs of personnel salaries [29,30]. Therefore, increasing the number of employees in teams in the absence of clear and hard scientific evidence for a significant increase in the quality of services provided would result in an unreasonable expenditure of funds and these resources are particularly limited in the healthcare sector.

The strength of the study was that the tests were performed in the same repeatable conditions, which influenced the credibility of the obtained measurements.

The study has a few limitations, among which it should be noted that several participants took part in both the teams of two and the teams of three. Another limitation was that a very high standard deviation was observed, which indicated a large diversity of the obtained results.

## 5. Conclusions

The analysis of the most examined components showed that rescuers in teams of three work more effectively than teams of two. 

However, this was not an unequivocal advantage of the three-person teams, which would justify the obligatory extension of the teams to three people. The widespread use of demonstrating in a system of two-person teams is a type of compromise, taking into account the satisfactory quality of services provided and the costs incurred.

The limited resources of the healthcare system indicate a low probability of introducing an obligatory increase in the number of personnel in emergency medical teams.

An improvement in the quality of the services provided could also be obtained by improving the quality of staff education without the need to increase expenditure on a larger number of staff.

## Figures and Tables

**Figure 1 ijerph-19-03753-f001:**
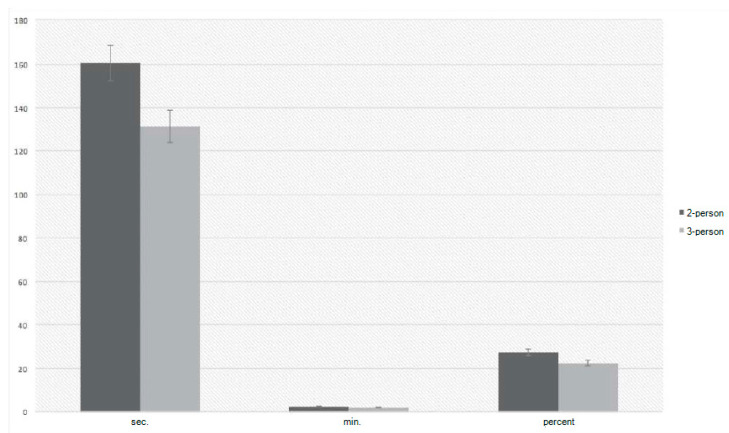
Comparison of chest compression interval variables in 2- and 3-person teams.

**Table 1 ijerph-19-03753-t001:** Differences in breaks in chest compressions, chest compression depth, frequency of chest compressions, electrotherapy-defibrillation rhythms, pharmacotherapy-non-defibrillation rhythms and oxygen therapy in teams of 2 and 3 people ^1^.

	2-Person Team (*n* = 100)		3-Person Team (*n* = 100)		t	*p*	95% CI		Cohen’s *d*
	M	SD	M	SD			LL	UL	
Seconds	160.42	42.56	131.19	37.76	5.14	<0.001	18.01	40.45	0.73
Percent	27.27	7.24	22.30	6.42	5.14	<0.001	3.06	6.88	0.73
Medium depth (mm)	45.38	8.04	48.04	6.04	−2.65	0.009	−4.64	−0.68	0.37
Chest compression frequency per minute	124.50	17.12	127.09	14.71	−1.15	0.253	−7.04	1.86	0.16
Time of first discharge	1.28	1.06	0.56	0.60	4.18	<0.001	0.38	1.06	0.84
Time of administration of the first epinephrine dose	2.56	1.03	1.79	0.77	4.24	<0.001	0.41	1.13	0.85
Time of rhythm notation	1.14	1.19	0.52	0.52	4.81	<0.001	0.37	0.88	0.68
Time of starting oxygen therapy	1.75	1.86	1.36	1.52	1.37	0.175	−0.17	0.94	0.23

^1^ M: mean; SD: standard deviation; 95% CI: confidence interval; LL and UL: lower and upper bounds of the confidence interval, respectively.

**Table 2 ijerph-19-03753-t002:** Cross-tabulation analysis of electrotherapy variables, pharmacotherapy variables, rhythm analysis variables and oxygen therapy variables in 2- and 3-person teams.

		Application of Gel	Gradation of Energy	Safety	BLS Whilst Charging Defibrillator	Defibrillation Delay	Analysis up to 2 min	Loading Every 2 min	Appropriate Dosage, Single Dosage	Dosage Every 3–5 min	Correct Recognition of the Rhythm	Quick Look	Electrodes	Analysis Every 2 min	Oxygen Therapy
2-Person Team															
No	N	43	39	8	33	17	32	33	0	27	3	64	3	62	44
	%	86.00	78.00	16.00	66.00	34.00	64.00	66.00	0.00	54.00	3.00	64.00	3.00	62.00	44.00
Yes	N	7	11	42	17	33	18	17	50	23	97	36	97	38	56
	%	14.00	22.00	84.00	34.00	66.00	36.00	34.00	100.00	46.00	97.00	36.00	97.00	38.00	56.00
3-Person Team															
No	N	34	36	7	25	36	13	19	2	11	0	33	1	25	11
	%	68.00	72.00	14.00	50.00	72.00	26.00	38.00	4.00	22.00	0.00	33.30	1.00	25.30	11.00
Yes	N	16	14	43	25	14	37	31	48	39	99	66	98	74	89
	%	32.00	28.00	86.00	50.00	28.00	74.00	62.00	96.00	78.00	100.00	66.70	99.00	74.70	89.00
X^2^		4.57	0.48	0.08	2.63	14.49	14.59	7.85	2.04	10.87	3.02	18.73	1.00	27.3	27.31
*p*		0.0032	0.488	0.779	0.105	<0.001	<0.001	0.005	2.04	0.001	0.082	<0.001	0.317	<0.001	<0.001
Vc		0.214	0.069	0.028	0.162	0.381	0.382	0.28	0.143	0.33	0.123	<0.001	0.071	0.37	0.37

## Data Availability

Data sharing not applicable.

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
