# Peer review of "Management Decisions: The Effectiveness and Size of the Emergency Medical Team"

_ijerph, 2022, doi:10.3390/ijerph19073753_

Round 1

Reviewer 1 Report

The document presents some important details that could limit its publication:
The introduction is very scarce, there is a lack of information in which the importance of this study is manifested.
The methodology does not indicate how the measurements were made for each variable under study and the statistical analyzes that were carried out are not mentioned.
In the results it is observed that the standard deviation is very high and even in one variable the standard deviation is larger than the mean.
In the discussion there is no comparison with other studies, which is why this section should be improved.
After reviewing the statistical analyzes you should rewrite the conclusion 

Author Response

Thank you for all suggestions. Please see the attachment.

Reviewer 2 Report

Thank you for the opportunity to review manuscript ID ijerph-1618126 entitled ‘Management decisions – effectiveness and the size of the emergency medical team’ which was submitted for potential publication to the International Journal of Environmental Research and Public Health.

This study reports the findings of a project analysing the impact of the size of the medical rescue team on resuscitation. On the whole I find the manuscript to be lacking in detail, particularly in the methods, and the presentation of the results to be confusing. The discussion needs more reflection on what the findings could mean for practice (and constraints around achieving best practice) and/or what research gaps remain. I look forward to reading a significantly revised manuscript and I wish the authors all the best in getting their study published.

General comments

All acronyms need to be explained at first use

Introduce what the ERC guidelines are in the introduction and then discuss how variables are assessed against these recommendations in the methods

Overall there are too many results tables. Where the same statistical test is being performed, I would suggest combining all variables being assessed into 1-2 tables max.

Do you have an assessment on impact of size of team on patient outcomes? That would be extremely powerful

Specific comments

Abstract

The abstract would benefit from a concluding sentence – what is recommended – 3 person teams? And is this likely to occur – what are the cost and staffing implications?

Introduction

Line 37 – doesn’t make sense as currently written – suggest ..’economic reasons often mean that’

Materials and Methods

Line 43 – I don’t understand the context or meaning of the first sentence of this section – suggest deleting or clarifying

I would move lines 44-48 to results to describe the participants – move this to the top of results before 3.1 heading

The methods need to be expanded – how did you recruit for the study – what was the promotion strategy?

Line 57 - Explain AMBU acronym

How was this data collected? – records, observations?

Results

How can there be 100 teams of 2 (200) and 100 teams of 3 (300) – were some of the 426 participants in both a 2 and 3 person team? Suggest clarifying at top of results.

Table 1 – I would assess in second or minutes (probably seconds) not both

Ensure all variables are listed in methods and described in the same way in the results – ie Breaks in chest compressions is not listed as a variable analysed in the methods, Compression depth is not listed as a variable in methods either

Mention assessment against ERC guidelines in methods and describe what these guidelines are. Also explain ERC acronym

Tables – assuming bolded result is significant but this is not stated anywhere – suggest adding & it has not been consistently done across all tables

Need to fix formatting of table 3

Type on table 4 header

Line 111 – who is they?

Discussion

I would like to see discussion about what this means for number of staff on teams and if guidelines could change or the constraints around that. What is needed to show 3-person teams provide better treatment and thus improve patient outcomes – is it more research? Or does a 2 person team not differ too much and a more economical way to provide care without impacting patient outcome? Even if you cant answer these questions it would be good to reflect on them

This study needs a strengths and limitations section

Conclusions

You need to tone down the claims made in the conclusions – how do we know this impacts patient outcomes/survival? What does this mean for practice. The current conclusion is too brief.

Author Response

Thank you for all suggestion. Please see the attachment.

Round 2

Reviewer 1 Report

The document has improved, however, there are some grammatical errors (line 110 and 122), it is recommended to review the entire document.

Author Response

Thank you for your suggestion. Please see the attachment.

Reviewer 2 Report

The paper has improved but is not ready for acceptance for publication yet. 

I would encourage the authors to improve the following components of the paper: 

The second last sentence in the abstract still does not make sense. Do you mean across all the elements or most of the elements - please clarify. 

There remains bolded findings in Table 1. 

The tables are condensed now which is great but still hard to read - they need to be reformatted to have the spacing reduced and perhaps organise n and % across the page rather than down to ensure the table can fit on one page to make it easier for the reader

I find the discussion to still be lacking. It is not about finding analogous research, more about reflecting on what your findings mean for practice. were there differences between 2 and 3 person teams? Do the differences warrant the expansion of teams (and costs associated with) from 2 person to 3. How likely is this to happen given 2 people teams were likely chosen due to cost of staffing for example. I would use this revised discussion to revise the concluding statement of the abstract which also does not make sense as currently written. 

Author Response

(The authors gave the same response as above.)
